# Image-Guided Liver Stereotactic Body Radiotherapy Using VMAT and Real-Time Adaptive Tumor Gating: Evaluation of the Efficacy and Toxicity for Hepatocellular Carcinoma

**DOI:** 10.3390/cancers13194853

**Published:** 2021-09-28

**Authors:** Marie Cantaloube, Florence Castan, Morgane Creoff, Jessica Prunaretty, Karl Bordeau, Morgan Michalet, Eric Assenat, Boris Guiu, Georges-Philippe Pageaux, Marc Ychou, Norbert Aillères, Pascal Fenoglietto, David Azria, Olivier Riou

**Affiliations:** 1Montpellier Cancer Institute (ICM), University Federation of Radiation Oncology of Mediterranean Occitanie, Montpellier University, INSERM U1194 IRCM, 34298 Montpellier, France; Marie.Cantaloube@icm.unicancer.fr (M.C.); morgane303@hotmail.com (M.C.); Jessica.Prunaretty@icm.unicancer.fr (J.P.); Karl.Bordeau@icm.unicancer.fr (K.B.); Morgan.Michalet@icm.unicancer.fr (M.M.); norbert.ailleres@icm.unicancer.fr (N.A.); pascal.fenoglietto@icm.unicancer.fr (P.F.); David.Azria@icm.unicancer.fr (D.A.); 2Biometrics Unit ICM, Montpellier Cancer Institute, University Montpellier, 34298 Montpellier, France; Florence.Castan@icm.unicancer.fr; 3Oncodoc, 34500 Béziers, France; 4Service d’Oncologie Médicale, CHU St Eloi, 34000 Montpellier, France; e-assenat@chu-montpellier.fr; 5Imagerie Médicale St Eloi, 34000 Montpellier, France; b-guiu@chu-montpellier.fr; 6Service d’Hépatogastroentérologie, CHU St Eloi, 34000 Montpellier, France; gp-pageaux@chu-montpellier.fr; 7Medical Oncology Department, Montpellier Cancer Institute (ICM), Montpellier University, INSERM U1194 IRCM, 34298 Montpellier, France; marc.ychou@icm.unicancer.fr

**Keywords:** stereotactic body radiation therapy, liver, hepatocellular carcinoma, VMAT

## Abstract

**Simple Summary:**

Although the use of stereotactic body radiation therapy (SBRT) in the management of hepatocellular carcinoma (HCC) remains unclear, it is a therapeutic option often considered in patients not eligible to or recurring after other local therapies. Liver SBRT can be delivered using a wide range of techniques and linear accelerators. We report the first evaluation for HCC of SBRT using volumetric modulated arc therapy (VMAT) and real-time adaptive tumor gating, which is a mainly completely non-invasive procedure (no fiducial markers for 65.2% of the patients). Our study showed that this SBRT technique has very favorable outcomes with optimal local control and a low toxicity rate.

**Abstract:**

Liver SBRT is a therapeutic option for the treatment of HCC in patients not eligible for other local therapies. We retrospectively report the outcomes of a cohort of consecutive patients treated with SBRT for HCC at the Montpellier Cancer Institute. Between March 2013 and December 2018, 66 patients were treated with image-guided liver SBRT using VMAT and real-time adaptive tumor gating in our institute. The main endpoints considered in this study were local control, disease-free survival, overall survival, and toxicity. The median follow-up was 16.8 months. About 66.7% had prior liver treatment. Most patients received 50 Gy in five fractions of 10 Gy. No patient had local recurrence. Overall survival and disease-free survival were, respectively, 83.9% and 46.7% at one year. In multivariate analysis, the diameter of the lesions was a significant prognostic factor associated with disease-free survival (HR = 2.57 (1.19–5.53) *p* = 0.02). Regarding overall survival, the volume of PTV was associated with lower overall survival (HR = 2.84 (1.14–7.08) *p* = 0.025). No grade 3 toxicity was observed. One patient developed a grade 4 gastric ulcer, despite the dose constraints being respected. Image-guided liver SBRT with VMAT is an effective and safe treatment in patients with inoperable HCC, even in heavily pre-treated patients. Further prospective evaluation will help to clarify the role of SBRT in the management of HCC patients.

## 1. Introduction

Hepatocellular carcinoma (HCC) is the sixth most common cancer in the world with an incidence of 854,000 new cases/year. In terms of mortality, it is the third most common cancer with 810,000 cases/year [1,2]. The incidence appears to be on the rise, in particular due to the increase in chronic hepatitis B and C, and the better diagnosis and management of underlying liver diseases [3]. HCC develops in 90% of cases in cirrhotic patients [4] and is one of the main causes of death. In Europe the main etiologies of cirrhosis are alcohol (50–75%), chronic viral hepatitis (15–25%), and non-alcoholic fatty liver disease (NASH) [5]. The preferred curative treatments for HCC is surgery resection, liver transplantation, and ablative hepatic radiofrequency (RF) [6]. Percutaneous radiofrequency (RF) destruction is an alternative when surgical treatment is contraindicated or liver transplantation is not possible [7,8]. However, the majority of patients are not eligible for curative treatment. In this case, trans arterial chemo embolization (TACE) is often performed in patients with preserved hepatic function, without extrahepatic invasion, or without ascites or portal thrombosis. For inoperable cancers, TACE has shown superiority in terms of overall survival compared to supportive care (16 months versus 20 months) [9]. The role of liver radiotherapy is currently under discussion. In fact, conformational radiotherapy and stereotactic body radiation therapy (SBRT) are not included as a curative therapeutic option in the various European, American, and French guidelines. Interestingly, various Asian guidelines and NCCN guidelines have added contents about SBRT to integrate this therapeutic option as an alternative to other local treatments for HCC patients [10,11,12,13]. To date, there are no randomized trials comparing radiotherapy with other local treatment options (RF and TACE). Historically, liver radiotherapy has been limited for a long time due to hepatic toxicity from total irradiation of the liver, especially with the occurrence of radiation induced liver disease (RILD) [14,15]. With the improvement of the precision of current techniques, in particular conformational radiotherapy allowing partial irradiation of the liver and SBRT allowing the delivery of high doses of radiation in small volumes, toxicities were reduced with less than 5% of RILD [16,17]. Non-randomized, prospective series published in the literature have shown the efficacy and safety of SBRT treatment [18,19,20,21,22,23,24], but its place in the therapeutic strategy for the management of HCC therefore remains to be defined. We retrospectively report the outcomes of a cohort of patients treated with SBRT in the treatment of HCC at the Institut du Cancer in Montpellier.

## 2. Materials and Methods

### 2.1. Population

The study was conducted according to the guidelines of the Declaration of Helsinki, and approved by our local ethics committee.

Our retrospective single-center study included 66 patients between March 2013 and December 2018 treated by liver SBRT at the Cancer Institute in Montpellier (ICM). The diagnosis of HCC was either histological or evaluated according to the Barcelona criteria (biological and radiological result). The patients were not eligible for further treatment or had a relapse after local therapy. The liver SBRT treatment was reviewed in multidisciplinary specialized tumor board with a liver surgeon, a hepatologist, a radiation oncologist, an oncologist specialized in digestive cancers, and a radiologist. The indication of SBRT was validated for inoperable patients (surgical contraindication due to the patient, or inoperable lesion due to its size or location), if there was a contraindication for RF or TACE or after failure of these treatments. The patients had to be in good general condition (Eastern Cooperative Oncology Group Performance Status (ECOG PS) 0–2), with compensated cirrhosis (≤Child B7), the absence of extrahepatic disease, or controlled extrahepatic disease. However, prior treatments such as RF, TACE, surgery, chemotherapy, or targeted therapy did not exclude them from our study.

### 2.2. Planification and Delivery of Treatment

Our methodology for Liver SBRT using VMAT and real-time adaptive tumor gating has already been described in detail previously [25,26,27,28]. Figure 1 gives an overview of our workflow for treatment planning and for adaptive treatment gating. Briefly, SBRT was performed in free breathing with respiratory control on a Truebeam^®^ Novalis accelerator (STX v 1.6) (Varian Medical Systems, Palo Alto, CA, USA) by dynamic arc therapy using 6MV photons. During irradiation, the IMR system (Intra Motion Review, Varian Medical Systems, Palo Alto, CA, USA) allows the acquisition of kV images synchronized to the RPM signal (Real Time Position Management System, Varian Medical Systems, Palo Alto, CA, USA) allowing setting and adapting a treatment window with an inhale and an exhale limit. Treatment was delivered one session per day, most often over five consecutive days. Noninvasive method and surrogate for tumor movement gating were preferred, for example liver dome kV and CBCT positioning. However, radiopaque fiducial markers (1 to 3 fiducial) were implanted percutaneously inside or near the HCC lesion, when necessary and in the absence of contraindications, at least one week before the simulation scan. This benchmark made it possible to follow the movement of the lesion during the respiratory cycle and therefore allowed tumor gating. The patient was simulated in the supine position, arms raised above the head, immobilized by a self-expanding foam mattress (Moldcare, Qualimedis, Maisons-Alfort, France). An RPM box was placed on the abdomen and recorded the respiratory signal. Three 4D simulation scans were performed, during the two weeks preceding the treatment, in free-breathing condition in order to ensure the reproducibility of the respiratory cycle. The dosimetric scanner included an injection of iodine with a tri-phase acquisition (arterial, portal and late phase). The 4D scanner consisted of ten respiratory phases (from 0 to 90%, every 10%). The thickness of the sections was 2.5 mm. Three reconstructions of the “Maximum Intensity Projection” (MIP), “Minimum Intensity Projection” (minIP) and “Average Intensity Projection” (Ave-IP) type were carried out. The isocenter was placed on the “Average-IP” series. A simulation 4D positron emission tomography (PET) scanner in free breathing, and a simulation MRI injected under physiological unforced inspiration and expiration conditions were performed in the absence of contraindication. The 4D PET scanner was performed with 18F-FDG or 18F-Choline depending on the case and availability [29,30]. The macroscopic tumor volume, i.e., the “Gross Tumor Volume” (GTV) was delineated on the simulation scanner. Co-registrations with the simulation MRI and the 4D PET scanner were performed to facilitate this delineation. The internal target volume, i.e., “Internal Tumor Volume” (ITV) was delineated on the MIP series and then on all the respiratory phases of the 4D scanner. The forecast target volume, i.e., the “Planning Target Volume” (PTV) was defined with an isocentric margin of 5 mm applied to the ITV. The fiducial delineation (on the MIP, the maximum inspiratory phase and the maximum expiratory phase), or of the hepatic dome in inspiratory and expiratory conditions, was performed. Organs at risk were delineated on the “average” series and included the liver, stomach, esophagus, duodenum, small intestine, spinal cord, kidneys, heart, ribs, biliary tract and lungs. The liver was delineated over all the respiratory phases in order to take into account, during dosimetry, its movement during respiration. The prescribed dose to the PTV was 50 Gy in five fractions. The choice of fractionation was adapted to hepatic function and to the dose volume histograms of organs at risk (OAR), the constraints of which are summarized in Table 1. The methods of monitoring the target lesion could be either kV of the hepatic dome (without implantation of fiducials), or kV with monitoring of fiducials. If it was not possible to place fiduciaries and lesions were far from the hepatic dome, only CBCTs were performed without using IMR.

### 2.3. Patient Follow-Up

The patients were assessed every three to four months after the end of treatment for the first year, then every four to six months. At each follow-up, clinical examination, contrast-enhanced MRI (or injected CT scanner) and blood sample including alpha-fetoprotein measure and liver function blood test. mRECIST criteria was used for tumor response evaluation. Response to treatment was classified as follows:Complete response if the hypervascularization or lesion had completely resolved;Partial response in case of reduction of at least 30% of the enhanced part of the target lesion or its size;Progression if there was an increase in size of at least 20% of the enhanced part of the target lesion, or in the size of the lesion or if a new lesion appeared;Stable if there was no argument for partial response or progression.

RILD was described according to Lawrence in classic or non-classic RILD. Classic RILD was defined as anicteric hepatomegaly associated with non-carcinomatous ascites and/or alkaline phosphatase greater than twice the upper normal limit or the value before treatment, occurring between two weeks and three months after irradiation. Non-classic RILD was defined as an elevation of transaminases greater than five times the upper limit of normal or a decline in the Child-Pugh score of two or more points in the absence of classical RILD.

### 2.4. Endpoints

Local control (LC) was defined as the absence of progression of the treated lesion. Survival without local recurrence is defined as the time between the radiation and the onset of local recurrence according to mRECIST criteria. Overall survival (OS) was defined as the time between the radiation and the date of death, regardless of the cause. Extra-target hepatic recurrence-free survival (ETHRFS) was defined as the time between the radiation and the extra-target hepatic recurrence. Metastasis-free survival (MFS) was defined as the time between the radiation and the diagnosis of metastases. Disease-free survival (DFS) was defined as the time between the radiation and the first event (local, hepatic, metastatic recurrence, death from whatever cause). Toxicity was graded according the Common Terminology Criteria for Adverse Events (CTCAE v.4)

### 2.5. Statistical Analysis

The quantitative variables were described by the number of observations (*n*), the median, the minimum and the maximum. The Kruskal–Wallis test was used for comparing the distributions of quantitative variables. The qualitative variables were described by the number of observations (*n*) and the frequency (%) of each of the modalities. The missing categories were counted. Fisher’s exact test was used for the comparison of proportions. Median follow-up was estimated using the inverse Kaplan–Meier method. The duration of the follow-up was defined by the time interval between the radiation treatment and the date of the last news, the death being censored. The Kaplan–Meier method was used for the analysis of survival data and for estimating median survival rates and times. In this case, the associated survival curves were presented. The proportional hazard Cox model was used to compare survival distributions after adjusting for possible prognostic factors, calculating hazard ratios (HR) and their 95% confidence intervals (CI). The contribution of each variable in the model was tested with the likelihood ratio test.

## 3. Results

### 3.1. Patients

The characteristics of the patients are shown in Table 2. Median follow-up was 16.8 months 95% CI (15.0–22.7). Most patients were men (94%). Median age at the time of inclusion was 69 years and most patients had an ECOG PS of 0 to 1 (%). The etiologies of cirrhosis were mainly alcoholic, alone in 23 patients, and associated with another etiology in 16 other patients. The second etiology was viral. Most patients had a Child A5 score (67%). One patient had an initial Child score greater than B7. The Barcelona clinic liver cancer (BCLC) score was predominantly 2 (*n* = 35), and the disease remained localized in 87% of patients (stage 1 and 2 TNM). There were four patients awaiting transplantation, of which two were transplanted (at four and seventeen months from the end of SBRT). Two-thirds of the patients (67%) received prior hepatic treatment (Table 3). Only 31 patients received no prior treatment for the lesion treated with SBRT. About 25% of the patients (*n* = 15) received at least two prior treatments.

### 3.2. Treatment

Among the 66 patients treated, 6 patients had two synchronous lesions, and the others had one. 72 lesions were treated. (Table 2). They were located mainly on the right liver. The dome lesions were localized in several hepatic segments, possibly in both the right and left liver. The median size of the lesions was 30 mm [range, 8–100 mm]. About 35% of patients had surgical clips or fiducials. Most patients (71%) received 50 Gy in 5 fractions of 10 Gy. The median total dose received at PTV was 50 Gy for a median volume of 79.5 cc. The median treatment time was seven days (range five to eighteen) All patients completed their treatment. More than half received 4D PET, mostly choline. This exam was considered useful for delineation in 61% of the cases (Table 3).

### 3.3. Local Control and Response

Local control was 100% at six months and one year. None of the patients treated with SBRT had a recurrence in the irradiated territory (in field) (Table 4). The objective response rate (complete response and partial response), according to the modified Response Evaluation Criteria in Solid Tumours (mRECIST) criteria was 50% at three months, 68.5% at six months, and 71% at one and two years (Table 4).

### 3.4. Survival Data

Survival data is described in Table 4 and Figure 2. The overall survival rate was 96.8% [95% CI, 87.9–99.2] and 83.9% [95% CI, 71.2–91.3] at 6 and 12 months, respectively. The median survival was 29.3 months [95%CI, 25.1 to not reached] A total of 19 patients died (28.9%). The progression-free survival rate was 76.4% [95%CI, 64.0–85.1] and 46.7% [95%CI, 33.2–59.1] at 6 and 12 months. The median progression-free survival was 10.6 months [95%CI, 8.8–15.1]. The extra-target recurrence-free survival rate (outfield) was 79.2% [95%CI, 66.9–87.4] and 58.1% [95%CI, 43.4–70.2] at 6 and 12 months, with a median of 15.1 months [95%CI, 10.5–38.0]. The survival rate without metastatic recurrence was 100% at 6 months and 97.7% [84.9–99.7] at 12 months.

### 3.5. Analysis of Prognostic Factors

The sum of the diameters of the lesions (HR = 2.62 (1.25–5.51) *p* = 0.018), the PTV (HR = 2.90 (1.48–5.68) *p* = 0.0035) and the number of prior treatments (≥2 treatments HR = 2.68 (1.24–5.80) *p* = 0.046) appeared as prognostic factors for disease-free survival in univariate analysis. The sum of the diameters of the lesions remains significantly associated with disease-free survival (HR = 2.57 (1.19–5.53) *p* = 0.02) in multivariate analysis (Table 5). The increased diameter of the lesions treated is therefore associated with a greater risk of relapse. The PTV (HR = 3.12 (1.17–8.33)) and the number of fractions (HR = 2.97 (1.09–8.07)) appeared as prognostic factors for overall survival during univariate analysis (Table 5). In multivariate analysis, only PTV remains associated with overall survival. No initial or treatment characteristic appeared to be related to the objective response. For local control, analysis of prognostic factors was not performed due to the absence of events.

### 3.6. Tolerance of Treatment

No patient experienced acute or late grade 3 toxicity (Table 6). A single grade 4 gastric ulcer was observed with a stomach Dmax of 29.55 Gy (which was in accordance to dose constraints). The treated lesion was located at the intersection of segments II and III, near the digestive structures. There were no deaths or discontinuation of treatment related to the treatment. At three months, four patients (6.9%) presented with decompensated cirrhosis. One patient (1.8%) presented with non-classical RILD with worsening of his child Pugh score by more than two points within three months of the end of radiotherapy. There have been no cases of classic RILD. At six months and at one year, one patient and six patients decompensated their cirrhosis. Three patients (5.2%) had a Child Pugh score ≥ B7 at three months, three patients at six months, and four patients at one year.

## 4. Discussion

### 4.1. Comparison of Our Results with Data from the Literature

Our study shows the feasibility, efficacy and safety of Liver SBRT using VMAT and real-time adaptive tumor gating for HCC lesions. The most frequently used dose was 50 Gy in 5 fractions of 10 Gy. The local efficacy was optimal since no local recurrence were reported and the objective response rate according to mRECIST criteria was 73% (95% CI [60.3–83.9]) with a complete response rate of 51.7%. Historically, the first study on liver SBRT published in 1995 by Blomgren et al. [34], showed a similar local control rate with 100% at 1 year and 95% at 2 years, but with excessive toxicity. Baumann et al. [35] in 2018 used a similar dose of 50 Gy in 5 fractions of 10Gy and found a local control rate at 1 year of 95%. Overall, the local control rates in retrospective studies vary between 87% and 100% at one year for treatment doses ranging from 30 to 60 Gy (Appendix A) [34,35,36,37,38,39,40,41,42,43,44,45,46,47,48,49,50]. We found seven published prospective studies (Appendix A). Bujold et al. [22] analyzed 102 patients treated with doses of 24 to 54 Gy (in 6 fractions) and showed a dose/response relationship for local control. Two predictive factors of local control were reported in three studies, namely the size of the tumor (less than or equal to 5 cm) and the treatment dose. A large meta-analysis showed that the local control for HCC SBRT was 87% and the overall survival at one year was 80%. The late toxicity rate was 6% [51]. Another more recent meta-analysis reported a two year local control rate of 84.5% [52]. In our study, due to lack of events we were unable to look for prognostic factors for local control.

Despite good local control, Disease-free survival in our study was 76.4% and 46.7% at six months and one year. These rates are consistent with those found in the literature since they vary between 41% and 90% at one year. These results may be explained by the existence in these patients of underlying liver disease and hepatic recurrence outfield with the need for retreatment. Overall PFS levels after SBRT are shorter than after TACE or RF, but patients have often been on multiple treatments and have failed several treatments at the time of treatment with SBRT. SBRT is often used late in the treatment course of HCC patients since the technique is newer and the evidence level poorer than other techniques. It might just reflect that more advanced patients are treated with SBRT. In our study, the prognostic factors influencing disease-free survival were the sum of the diameter of the lesions (multivariate analysis) and the number of prior treatments (univariate analysis). These low PFS levels rise the potential of combining local therapy with systemic treatment. Regarding overall survival, the rate was 96.8% at six months and 83.9% at one year, with a median survival of 29.3 months. Reported overall survival rates vary between 55 and 100% at one year. Heterogeneous analysis populations, particularly in terms of disease stage, size and volume of treated lesions, could explained differences in overall survival. Yeung et al. [49] found in univariate analysis that small lesions were associated with better overall survival. Likewise, a BED > 100Gy would be a prognostic factor for overall survival. Scorsetti et al. [53] showed a significant association between local control and overall survival with a median survival of 18.8 months in controlled patients compared to 7.8 months in uncontrolled patients (*p* < 0.04). Several predictive factors for survival have been identified: general condition, diagnostic status of hepatocellular carcinoma (initial non-recurrent), tumor size, total treatment dose and tumor response. In our study, the volume of PTV was a prognostic factor for overall survival, in multivariate analysis.

The tolerance of the treatment was excellent. We had only one case of non-classical RILD (by Lawrence’s criteria), and no toxic deaths. We didn’t report any discontinuation of treatment for toxicity. During follow-up, only one patient presented with a grade 4 peptic ulcer. The implication of SBRT in the occurrence of this event is plausible due to the site of the lesion treated. The dose constraints for the digestive organs were nevertheless respected. Our results compared favorably with those found in the literature. In fact, gastrointestinal toxicities of grade greater than 2 vary between 1% and 12% and the rates of RILD between 2% and 17.6% (Appendix A). Debbi et al. [54] have shown that hepatic toxicity was related to the liver volume spared, the size of the lesion treated, the irradiation technique, the number of fractions, the initial hepatic function, the nature of the lesion treated. In our study, the dose constraints to the liver were all respected.

### 4.2. SBRT and Prior Treatments

More than half of our patients (66%) had received prior treatment, mostly RF or TACE. However, despite these previous local treatments, the local control rates of SBRT were excellent. Hepatic SBRT can be used as salvage therapy for previously multi-treated patients, in case of local failure or in the event of hepatic evolution, with good tolerance and efficacy. In our study, the number of prior treatments was associated with poorer disease-free survival in univariate analysis. Thanks to the low toxicity rate, it does not prevent further treatment, especially further SBRT, in the event of extra target recurrence. This was the case for six of our patients, including one who received two additional SBRT treatments, with good local control each time.

### 4.3. Limits and Bias

Our study is limited by its retrospective and monocentric nature with small sample size. However, it is one of the largest series specifically addressing SBRT for HCC and the first one evaluating VMAT and real-time adaptive tumor gating. It is necessary to collect as many experiences as possible while waiting for large prospective studies to be published. STEREOLIVER (ClinicalTrials.gov Identifier: NCT03408665) is a French large multicentric phase two prospective study on liver SBRT which hopefully will give robust data for HCC patients.

### 4.4. SBRT as a Bridge Therapy

Several studies have suggested that SBRT plays a role as a bridge therapy to transplantation, as it is well tolerated and may result in a complete pathological response [55,56]. Among our study population, four patients were awaiting liver transplantation and two underwent liver transplantation after receiving SBRT.

### 4.5. The Role of Liver SBRT for HCC

There are no randomized, comparative studies comparing SBRT with other treatment techniques, such as RF or TACE. Compared to other minimally invasive procedures, SBRT would offer at least comparable or even more favorable efficacy for stages I to III or patients with CHC BCLC A or B. In a retrospective study of 244 patients with inoperable HCC, Wahl et al. [57] compared radiofrequency ablation (RFA) with SBRT and reported similar two-year local control rates (80.2% vs. 83.8%). Lee et al. compared RFA and SBRT in a recent meta-analysis and showed that the pooled two-year local control rate was not significantly different between SBRT and RFA, although favoring SBRT (84.5% vs. 79.5%, respectively). However, the pooled analysis of overall survival (OS) in HCC studies showed an odds ratio of 1.43 (95% CI: 1.05–1.95, *p* = 0.023), favoring RFA [52].

The American Association for the Study of Liver Diseases (AASLD) and the European Organization for research and treatment of cancer (EORTC), do not consider liver SBRT in the management of HCC. It is not a curative therapeutic option according to the EASL (European Association for the study of the Liver Practical). French HAS (Haute Autorité de Santé) does not consider SBRT to be a standard, despite encouraging results. Only the National Comprehensive Cancer Network (NCCN) [58] has integrated conformational or stereotactic radiotherapy into treatment algorithms, as an option in inoperable tumors. The SFRO (French radiotherapy society) has established selection criteria for liver SBRT which are a number of lesions less than or equal to three, a tumor size not exceeding 5 cm and a volume of healthy liver parenchyma of at least 700 cm^3^ [16,59,60]. Added to this is a preserved general condition, a Child Pugh ≤ B7 and controlled extrahepatic disease. Portal thrombosis or ascites are not contraindications to performing SBRT, unlike other local techniques. In all cases, the indications for liver SBRT should be validated collegially in specific and specialized multidisciplinary staff.

## 5. Conclusions

Our study shows the feasibility and the efficacy in terms of local control with the absence of recurrence on the irradiated site and the safety of liver SBRT using VMAT and real-time adaptive tumor gating for HCC lesions, including patients who have received one or more prior treatments. The results are consistent with those published so far. More prospective data is necessary to compare the different SBRT techniques and to better establish its place for the management of HCC.

## Figures and Tables

**Figure 1 cancers-13-04853-f001:**
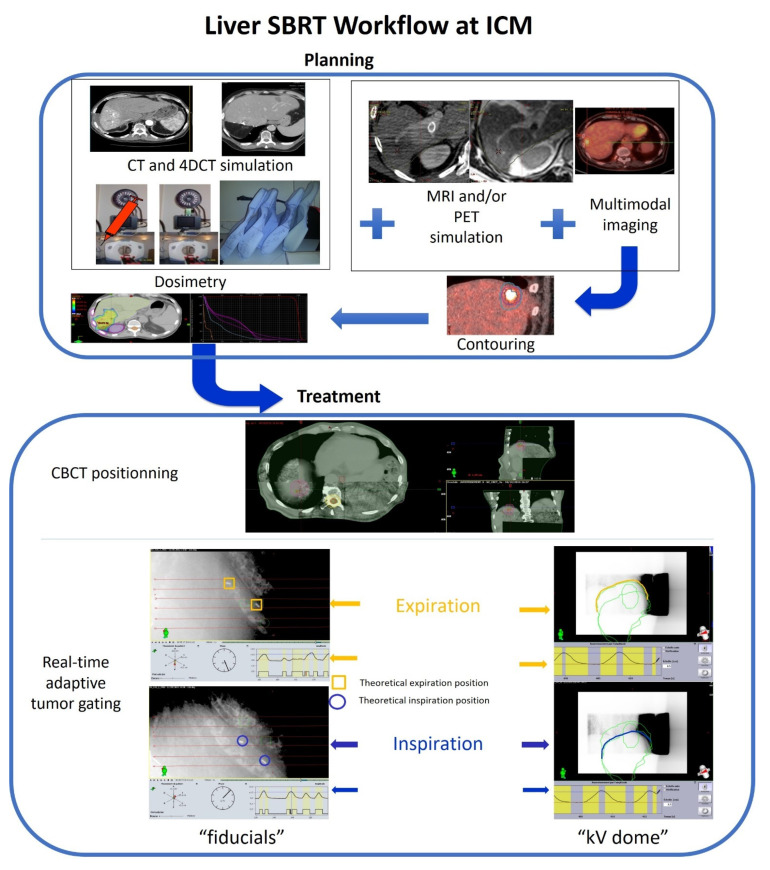
Liver SBRT workflow at ICM.

**Figure 2 cancers-13-04853-f002:**
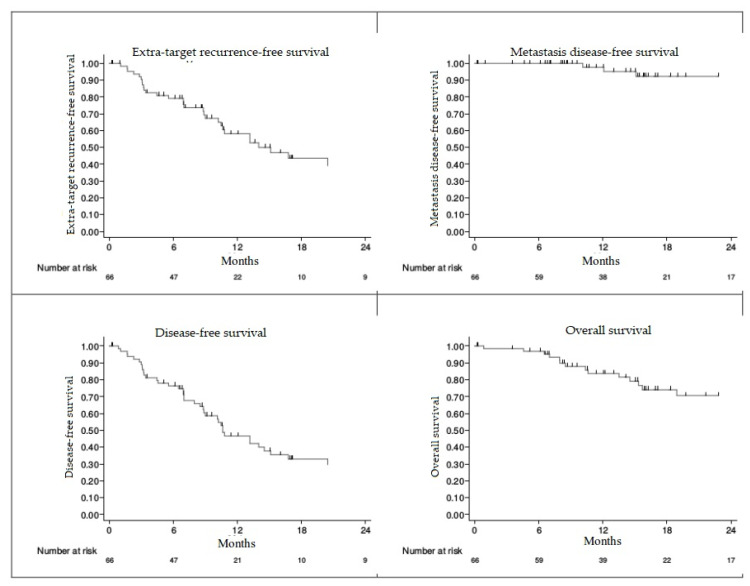
Survival Curves—Kaplan Meier.

**Table 1 cancers-13-04853-t001:** Dose constraints at OARs [31,32,33].

OAR	Constraints (5 Fractions)	
	Mean/Max Dose	Dose-Volume
Liver-PTV		700 cc or more receiving 0 to 15 Gy (if liver > 2000 cc) or 35% or more receiving 0 to 15 Gy
		(if liver < 2000 cc)
Liver-GTV		700 cc or more receiving 0 to 15 Gy (if liver > 2000 cc) or 35% or more receiving 0 to 15 Gy (if liver < 2000 cc)

Liver-GTV	D_mean_ < 18 Gy	
(>5 fractions)		
Kidneys	Lowest D_mean_	V_18_ ≤ 33%
Spinal cord	D_max_ ≤ 25 Gy	V_22.5_ ≤ 0.25 cc
		V_13.5_ ≤ 1.2 cc
Heart	D_max_ ≤ 35–38 Gy	V_32_ ≤ 15 cc
Stomach	D_max_ ≤ 31 Gy	V_28_ ≤ 10 cc
Duodenum	D_max_ ≤ 31 Gy	V_18_ ≤ 5 cc
Small bowel	D_max_ ≤ 29 Gy	V_19.5_ ≤ 5 cc
Colon	D_max_ ≤ 29 Gy	V_25_ ≤ 5 cc
Esophagus	D_max_ ≤ 35 Gy	V_19.5/27.5_ ≤ 5 cc
Ribs	D_max_ ≤ 43 Gy	V_135_ ≤ 1 cc

**Table 2 cancers-13-04853-t002:** Characteristics.

Parameter		*n* (%)	Median [Min–Max]
Patients Characteristics			
Gender	Women	4 (6.1%)	
	Men	62 (93.9%)	
Age (years)		66	69 (51–84)
ECOG	0	16 (24.2%)	
	1	46 (69.7%)	
	2	4 (6.1%)	
Etiology of cirrhosis	No cirrhosis	5 (7.6%)	
	Alcoholic	23 (34.9%)	
	Viral	10 (15.1%)	
	Mixed	17 (25.8%)	
	NASH	3 (4.5%)	
	Other *	3 (4.5%)	
	Unknown	5 (7.6%)	
ChildPughscore	A5	44 (67.7%)	
	A6	16 (24.6%)	
	B7	4 (6.2%)	
	>B7	1 (1.5%)	
	Missing	1	
BCLC score	1	2 (5%)	
	2	35 (87.5%)	
	3	3 (7.5%)	
	Missing	26	
Alpha fetoprotein value	Yes No	60 (90.9%) 6 (8.1%)	6.6 (0.6–4000)
Prior Treatments			
Previous treatment for HCC	Yes	44 (66.7%)	
	No	22 (33.3%)	
Previous treatment for targeted lesion	None	31(47%)	
	1	20(30.3%)	
	≥2	15(22.7%)	
	Surgery	2 (3%)	
	RF	6 (9.1%)	
	TACE	10 (15.2%)	
	RF + TACE	6 (9.1%)	
	Other **	11 (16.6%)	
Waiting for transplant	Yes	4 (6.1%)	
	No	62 (93.9%)	
Treated Lesions			
Number of lesions	1	60 (90%)	
	2	6 (10%)	
Diameter of the lesion (mm)		66	30 (8–100)
Location of the lesion	Dome	9 (13.6%)	
	Segment 1	4 (6.1%)	
	Right side	31 (47%)	
	Left side	16 (24.2%)	
	Others	6 (9.1%)	

* other = Hemochromatosis, overlap syndrome and cholangitis; ** other = alcoholization, targeted therapy, association of treatment (TACE + sorafenib, Surgery + RF, chemotherapy + sorafenib).

**Table 3 cancers-13-04853-t003:** Dosimetric parameters.

Parameters	Subgroup	*n* (%)	Median [Min–Max]
Surgical clips	No	43 (65.2%)	
	Yes	23 (34.8%)	
Method used to monitor the lesion	kV dome	12 (18.2%)	
	CBCT only	26 (39.4%)	
	kV dome + CBCT	5 (7.6%)	
	Other (fiducials, surgical clips, prosthesis, etc.)	23 (34.8%)	
4D PET	No	30 (45.5%)	
	Yes	36 (54.5%)	
PET type	Choline	24 (66.7%)	
	18-FDG	12 (33.3%)	
Lesion fixation	No	13 (36.1%)	
	Yes	23 (63.9%)	
Usefulness for contouring	No	14 (38.9%)	
	Yes	22 (61.1%)	
Healthy liver	Liver volume (cc)	66 (100%	1462 (850–3261)
	Healthy liver volume (cc)	66 (100%)	1367 (599–2967)
	PTV/healthy liver ratio	66 (100%)	0.1 (0.1–2)
	D_mean_ (Gy)	66 (100%)	9.6 (.3–17.5)
	V < 15 Gy (cc)	66 (100%)	1109.6 (65.1–2446.3)
	(V_healthy liver_—V_15_)		
	V_15_ (cc)	66 (100%)	297 (52–1206.7)
Target volumes (cc)	PTV	66 (100%)	79.5 (13.3–722.7)
	GTV1	3 (4.5%)	27.5 (1.5–463.4)
	GTV2	3 (4.5%)	5.9 (4.4–19.7)
	ITV1	64 (97%)	30 (2.6–485)
	ITV2	3 (4.5%)	7.5 (5.1–39.6)
Treatment scheme	50 Gy—5 fractions	44 (71%)	
	50 Gy—10 fractions	7 (11.3%)	
	41 Gy—10 fractions	4 (6.5%)	
	44 Gy—10 fractions	1 (1.6%)	
	45 Gy—5 fractions	3 (4.8%)	
	45 Gy—10 fractions	1 (1.6%)	
	40 Gy—10 fractions	1 (1.6%)	
	36 Gy—10 fractions	1 (1.6%)	
Dose (Gy)	Total dose to PTV	66 (100%)	50 (35–50)
	Dose per fraction	66 (100%)	10 (3.5–10)
Staggering (days)		66 (100%)	7 (5–18)

**Table 4 cancers-13-04853-t004:** Survival data and response rate to radiotherapy according to mRECIST criteria.

Survival Data								
Survival parameter			Survival Rate (IC 95%)				Median Survival	
			6 Months		12 Months		(IC 95%)	
Without local recurrence			100%		100%		/	
(0 event)			/		/		/	
Without extra-target hepatic			79.2%		58.1%		15.1 months	
recurrence (31 events)			(66.9–87.4)		(43.4–70.2)		(10.5–38.0)	
Without metastatic			100%		97.7%		/	
recurrence (3 events)			/		(84.9–99.7)		/	
Disease free			76.4%		46.7%		10.6 months	
(41 events)			(64.0–85.1)		(33.2–59.1)		(8.8–15.1)	
Overall survival			96.8%		83.9%		29.3 months	
(19 events)			(87.9–99.2)		(71.2–91.3)		(25.1-not reached)	
**Response Rate (mRECIST Criteria)**								
Response	3 months (*n* = 59)	6 months (*n* = 54)		12 months (*n* = 35)	24 months (*n* = 14)	36 months (*n* = 3)		Better response
				*n* (%)				*n* (%) (IC 95%)
**Objective**	**29 (50%)**	**37 (68.5%)**		**25 (71.4%)**	**10 (71.4%)**	**2 (66.7%)**		**44 (73.3%) (60.3–83.9)**
Complete	16 (27.6%)	23 (42.6%)		18 (51.4%)	10 (71.4%)	2 (66.7%)		31 (51.7%)
Partial	13 (22.4%)	14 (25.9%)		7 (20%)	0	0		13 (21.7%)
Stability	23 (39.7%)	11 (20.4%)		1 (2.9%)	0	0		12 (20%)
Progression	6 (10.3%)	6 (11.1%)		9 (25.7%)	4 (28.6%)	1 (33.3%)		4 (6.7%)
Missing	1	0		0	0	0		0

Objective response = Complete response + partial response.

**Table 5 cancers-13-04853-t005:** Overall survival (univariate Cox model) and disease-free survival (multivariate Cox).

Overall Survival (Univariate Cox Model)										
Parameter	Subgroup		N Event/ N Total	1-Year OS (%)		HR		(95% CI)	*p* Value *	
Age			19/66			0.95		(0.89–1.01)		
	<70 years		12/40	82.8%		1			0.1	
	≥70 years		7/26	85.9%		0.7		(0.27–1.85)	0.47	
Sum of lesion diameters									0.16	
	<50 mm		14/54	84.6%		1				
	≥50 mm		5/12	80%		2.23		(0.77–6.43)		
PTV (cc)	<129		8/44	90.9%		1			0.025	
	≥129		11/22	71.3%		2.84		(1.14–7.08)		
	<150		11/49	87%		1			0.027	
	≥150		8/17	75.3%		3.12		(1.17–8.33)		
Total Dose	<50 Gy		3/15	79%		1			0.9	
	≥50 Gy		16/51	85.5%		1.08		(0.31–3.78)		
Dose per fraction	<10 Gy		7/21	72.2%		1			0.19	
	≥10 Gy		12/45	89.4%		0.52		(0.20–1.35)		
Number of fractions	5		12/49	90.2%		1			0.04	
	10		7/17	65.6%		2.97		(1.09–8.07)		
Disease-Free Survival (Multivariate Cox)										
*n* = 66, events = 41		Subgroup			HR		(95% CI)			*p* Value *
Sum of lesion diameters		<50mm			1					0.02
		≥50 mm			2.57		(1.19–5.53)			
Previous treatment		None			1					0.059
		1			1.4		(0.65–3.01)			
		≥2			2.61		(1.20–5.66)			

* likelihood ratio test.

**Table 6 cancers-13-04853-t006:** Tolerance of SBRT.

Complication	Grade	3 Months (*n* = 59)	6 Months (*n* = 54)	12 Months (*n* = 35)	24 Months (*n* = 15)	36 Months (*n* = 3)
		*n* (%)				
Duodenal ulcer	Grade 0	57 (96.6%)	53 (100%)	29 (100%)	14 (100%)	3 (100%)
	Grade 2	2 (3.4%)	0	0	0	0
	Missing	0	1	6	1	0
Gastric ulcer	Grade 0	58 (98.3%)	53 (100%)	29 (100%)	14 (100%)	3 (100%)
	Grade 4	1 (1.7%)	0	0	0	0
	Missing	0	1	6	1	0
Hepatitis	Grade 0	58 (98.3%)	52 (100%)	28 (96.6%)	14 (100%)	3 (100%)
	Grade 2	1 (1.7%)	0	1 (1.34%)	0	0
	Missing	0	2	5	1	0
Nausea	Grade 0	58 (98.3%)	52 (100%)	29 (100%)	14 (100%)	3 (100%)
	Grade 2	1 (1.7%)	0	0	0	0
	Missing	0	2	6	1	0
Vomiting	Grade 0	59 (100%)	52 (100%)	29 (100%)	14 (100%)	3 (100%)
	Missing	0	2	6	1	0
Asthenia	Grade 0	46 (78%)	43 (82.7%)	21 (72.5%)	7 (53.9%)	3 (100%)
	Grade 1	13 (22%)	8 (15.4%)	5 (17.2%)	6 (46.1%)	0
	Grade 2	0	1 (1.9%)	3 (10.3%)	0	0
	Missing	0	2	6	2	0
Diarrhea	Grade 0	59 (100%)	52 (100%)	29 (100%)	14 (100%)	3 (100%)
	Missing	0	2	6	1	0
Ascite	Grade 0	54 (91.5%)	51 (98.1%)	24 (82.8%)	11 (73.3%)	3 (100%)
	Grade 1	5 (8.5%)	1 (1.9%)	3 (10.3%)	4 (26.7%)	0
	Grade 2	0	0	2 (6.9%)	0	0
	Missing	0	2	6	0	0
Classic RILD	No	56 (100%)	52 (100%)	29 (100%)	13 (100%)	3 (100%)
	Missing	3	2	6	2	0
Non-classic RILD	No	55 (98.2%)	52 (100%)	28 (100%)	13 (100%)	3 (100%)
	Yes	1 (1.8%)	0	0	0	0
	Missing	3	2	7	2	0

## Data Availability

Data available upon request.

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
