# Peer review of "Image-Guided Liver Stereotactic Body Radiotherapy Using VMAT and Real-Time Adaptive Tumor Gating: Evaluation of the Efficacy and Toxicity for Hepatocellular Carcinoma"

_cancers, 2021, doi:10.3390/cancers13194853_

Round 1

Reviewer 1 Report

In their manuscript " Image-guided liver stereotactic body radiotherapy using VMAT and real-time adaptive tumor gating : Evaluation of the efficacy and toxicity for hepatocellular carcinoma", authors presented an interesting retrospective study that reports the outcomes of HCC patients treated with SBRT.
The paper is well written, presents interesting results, and perfectly describes the evidence and limitations of the study.
My only comment will be related to the form, I suggest that the terms used as abbreviations be well defined when they first appear (ECOG, BCLC score, ...). This will enable the non-clinician reader to understand the data more easily.

Reviewer 2 Report

The authors should be commended for their retrospective analysis of treating HCC with SBRT. As the authors aptly state, SBRT is more frequently employed as a later treatment approach after others have been exhausted. This manuscript describes a cohort of 66 patients and describes very thoroughly the various clinicopathologic, treatment set-up, dosimetric, outcome and toxicity parameters. This is an important contribution to the growing literature demonstrating efficacy of SBRT in HCC.

Minor concerns: various English grammatical errors through manuscript. Some example replacements are below:

-Simple summary: Though the use of SBRT… (line 16)

line-35 – “developed” a grade 4

line 42 – “sixth most common”

line 44- “third most common”

line 68 – “the role of liver”

line 71 – “randomized trials”

line 89 – “reviewed in multidisciplinary”

line 127 = “arterial phase due to”

line 143 – “monitoring of fiducials”

line 312 – “PFS levels support the potential”

Table 8 – Disease-free

Reviewer 3 Report

This study has its value of reporting clinical experience of SBRT for HCC from Western country, which has been relatively rarely reported. I have some revisions to support acceptance.

Introduction: SBRT for HCC has been actively studied recently. I recommend authors to focus more about the subject of SBRT for HCC , not HCC treatment in general. Furthermore, many Asian guidelines (from Korea, Hong Kong, China) had contents about SBRT. It is good to mention those.

Method 2.2: Authors’ institution is using very well equipped and updated methodology of respiratory gating. It will make the manuscript much better if authors can add a related figure.

Page 6 to 8: It is not recommended to make three tables for patient characteristics. Please merge into a single table. Furthermore, there are too many staging information. I think BCLC and CPC score is enough. Rather, please add covariate of vascular involvement (e.g. portal vein thrombosis or inferior vena cava involvement). Please mention hemochromatosis, overlap syndrome, cholangitis as into ‘others’.

AFP ‘dosage’ is it rightly written?

Table 4: no1 no2 no1+2 <= difficult to understand

Table 6,9: Global survival => overall survival

Merge Table 6 and 7

Merge table 8 and 9

Line 299: I think Ref 47 in line 299 is miswritten. This result is referring study with PMID 30773180.

Discussion 4.2, 4.4; These are too lengthy although they are not related to the main subject. Please make it very brief.

Discussion 4.5.: Since this is not a review article, explanation of all references of accessory subject is unnecessary. I think brief explanation of Wahl study (54) and Lee meta-analysis (47) is enough.

Round 2

Reviewer 3 Report

It is so much nicer manuscript now.

Thank you for giving me opportunity to review this article.

I fully support acceptance now. I also suggest the authors to review manuscript carefully during proofreading, since many parts are changed.